# An Overview of Physical Exercise Program Protocols and Effects on the Physical Function in Multiple Sclerosis: An Umbrella Review

**DOI:** 10.3390/jfmk8040154

**Published:** 2023-11-04

**Authors:** Martina Sortino, Luca Petrigna, Bruno Trovato, Alessandra Amato, Alessandro Castorina, Velia D’Agata, Grazia Maugeri, Giuseppe Musumeci

**Affiliations:** 1Department of Biomedical and Biotechnological Sciences, Section of Anatomy, Histology and Movement Science, School of Medicine, University of Catania, Via S. Sofia 97, 95123 Catania, Italy; martina.sortino@unict.it (M.S.); luca.petrigna@unict.it (L.P.); bruno.trovato@phd.unict.it (B.T.); alessandra.amato@unict.it (A.A.); vdagata@unict.it (V.D.); graziamaugeri@unict.it (G.M.); 2Laboratory of Cellular and Molecular Neuroscience (LCMN), School of Life Sciences, Faculty of Science, University of Technology Sydney, Sydney, NSW 2007, Australia; alessandro.castorina@uts.edu.au

**Keywords:** MS, exercise, movement, exercise training

## Abstract

Multiple sclerosis is a disease that concerns a growing number of people, especially females. There are different interventions proposed for this population, and physical activity is one of them. A proper and well-structured physical activity program can be a cheap, feasible, and practical instrument to help this population improve their quality of life. Consequently, the present study aimed to analyze, through an umbrella review, published articles to evaluate the protocols and the effect of intervention on different types of multiple sclerosis and eventually to propose a standardized intervention for this population. Systematic reviews and meta-analyses of randomized controlled trials on multiple sclerosis and physical activity effects were searched for on the electronic databases PubMed, Web of Science, and Scopus up to 22 December 2022. The quality of the studies included was determined and the results were narratively analyzed. The included studies present heterogeneity in the population, in the study design and protocols, and in the outcomes evaluated. Most of the studies detected positive outcomes on the physical function of people with multiple sclerosis. This study highlights the necessity of future studies on a population with similar characteristics, adopting similar protocols to evaluate their feasibility and validity to make physical intervention prescribed as a medicine.

## 1. Introduction

Multiple sclerosis (MS) is an immune-mediated disease that affects the central nervous system and is one of the most prevalent neurological conditions and leading causes of impairment in young adults [1]. It was stated that there would be about 2.8 million people with MS worldwide by 2020, and females are twice as likely to have the disease as males [2]. This could be due to the higher survival rate among the female population [3]. MS is characterized by inflammatory demyelination with disruption to terminal axonal structures [4]. It leads to irreversible neurological damage [4]. MS is considered a two-stage disease that starts with early inflammation, which is the cause of the relapsing-remitting form, and delayed neurodegeneration, which is the cause of the progression of the non-relapsing form [5]. The most frequent form of MS is relapsing-remitting, characterized by the onset of recurrent clinical symptoms followed by total or partial recovery; moreover, after 10 to 15 years, a stage of the disease defined as secondary progressive MS causes progressive degeneration over time, worsening the clinical symptoms [6]. On the other hand, in the form of primary progressive MS, there is a gradual deterioration from the onset and the disease progression is unstoppable [7].

The pharmacological treatment of MS targets acute attacks, reduction in symptoms, and biological activity [8]. Among the most common symptoms are depression, pain, and walking difficulties [9]. There is also fatigue, usually perceived fatigue, which limits the activities of daily living and impacts anxiety, depression, cognition, and sleep quality, reducing the quality of life [10,11]. Pain, gait dysfunction, and fatigue, in that order, influence the perceived health of people with MS [12]. Most of the common symptoms can lead to a lower quality of life and activity restrictions, and they include cognitive deficiencies and muscular stiffness [13]. Related to muscle problems, it has been detected that people with MS generally present leg weakness, limiting their performance [14], especially their walking performance. The walking ability of this population is reduced in terms of velocity [15] but also in terms of cadence and stride length [16]. Postural balance control capacity is also reduced [16], limiting the daily activities and further worsening quality of life. This creates a circle that deteriorates the person’s state of health. Drugs that target the immunological signaling proteins or the immune cell populations are often adopted to treat MS; however, these treatments do not cure all of the symptoms of the disease but rather mainly decrease inflammation in these patients [17].

Exercise training has emerged as a useful rehabilitation strategy to control symptoms, regain function, improve quality of life, promote well-being, and increase involvement in activities of daily living [18]. Physical training has an important role in reducing perceived fatigue [19,20], and this indirectly improves quality of life. However, different training methods seem to have different effects. For example, the effects of aerobic exercise can be cited as leading to an improvement in the satisfaction of MS patients with their physical, mental, and social functioning [21]. On the other hand, resistance training seems to positively influence the production of neurotrophies and thus indirectly limit the progression of the disease [22]. Also, mindfulness training such as yoga seems to improve postural balance, speed, and endurance for walking, reducing fatigue, stress, anxiety, and depression and improving quality of life [23,24]. In summary, in people with MS, a supervised and customized exercise program can improve physical fitness, functional capacity, quality of life, cognitive impairments [25], aerobic capacity, and muscular strength, and it may improve mobility, fatigue, and health-related quality of life [26].

Neurologists, advanced practice clinicians, and other medical professionals can recommend physical activity and exercise, highlighting the advantages of treating the symptoms, increasing general health and quality of life, and motivating their patients throughout the treatment [27]. The common goal should be to delay or avoid irreversible neurological damage and maximize self-sufficiency, especially in activities of daily living. People with MS are generally less physically active, highlighting the necessity of proper and adapted intervention to improve patients’ adherence and compliance [28].

A standard operating procedure is a step-by-step description of the intervention to improving its quality and allow for repetition [29]. Thousands of articles are published each year on the topic of MS and physical activity or exercise; most of them are heterogeneous in terms of population and intervention. Therefore, this umbrella review evaluated previously published systematic reviews and meta-analyses of randomized controlled trials on the same topic [30] to evaluate the protocols adopted and their effects on each different type of MS and, eventually, to propose a detailed intervention for this population. The investigation of physical activity’s impacts on MS and the extrapolation of information about exercise training were also taken into consideration.

## 2. Materials and Methods

This umbrella review followed the Preferred Reporting Items for Systematic Reviews and Meta-Analyses (PRISMA) guidelines [31].

### 2.1. Search Strategy

The search for relevant articles was conducted on the electronic databases PubMed, Web of Science, and Scopus. The inclusion criteria for the articles were systematic reviews and meta-analyses published up to 22 December 2022. The search used various keywords, including “multiple sclerosis”, “MS”, “exercise”, “exercise training”, “physical activity”, “review”, and “meta-analysis”. The keywords were combined using the Boolean operators AND or OR. To search the three databases, a string was used: (“multiple sclerosis” OR “MS”) AND (exercise OR “physical activity” OR “exercise training”) AND (“systematic review” OR “meta-analysis”).

### 2.2. Eligibility Criteria

The inclusion and exclusion criteria for the population, intervention, comparison, outcomes, and study design (PICO-S) were carefully considered. The population under investigation was individuals with MS, regardless of age and MS typology. Reviews were excluded if the sample investigated included other concomitant pathologies. Studies were excluded if the physical exercise interventions were not structured and presented. The intervention had to include physical exercise. The comparison and the outcomes were not necessary because our focus was on the intervention’s structures. Other studies designed differently than systematic reviews and meta-analyses of randomized controlled trials were excluded. Only articles written in the English language were included regardless of the country of publication.

### 2.3. Data Sources, Study Sections, and Data Extraction

In the first step, manuscripts were stored in EndNote X8 (EndNote version X8; Thompson Reuters, New York, NY, USA), and duplicate selection was performed. In the second phase, two independent investigators screened the reviews against the eligibility criteria based on the title, abstract, and full text. Any disagreements between the two investigators were resolved by the principal investigator.

Information related to the first author and year of publication, review methodology, databases screened, number of reviews included, objective of the study, risk of bias assessment and score, conclusion of the study, population screened, training characteristics, and main results were stored in tables. A descriptive and narrative synthesis was adopted to describe the results. A meta-analysis was not performed due to the possibility of including studies considered in more than one systematic review, which increases the risk of bias [32].

### 2.4. Quality Assessment

The quality of the included systematic reviews and meta-analyses of randomized controlled trials was assessed using the rating scale “Assessment of Multiple Systematic Reviews” (AMSTAR) [33]. This scale comprises 11 items and has demonstrated reliability and validity [34]. Studies with a final score between 0 and 4 were considered of poor quality, those with a score between 5 and 7 were considered of moderate quality, and those with a score above 8 were considered of high quality. A score of 0 was assigned if no sufficient information was available, and a score of 1 was assigned if enough information was collected. All included reviews were independently scored by two investigators, and any disagreements were resolved by the principal investigator.

## 3. Results

A total of 1561 studies (PubMed 626; Web of Science 413; Scopus 522) were found after the search of the electronic database. After the removal of duplicate articles, 1099 remained. After title and abstract screening, a total of 65 studies were collected for full-text analysis. A final total of 16 systematic reviews and meta-analyses of randomized controlled trials were included in this umbrella review. The screening process is summarized in Figure 1.

### 3.1. Characteristics of the Included Studies

Fourteen (of 16) studies adopted PRISMA guidelines, while in two studies information was not provided. The minimum number of databases searched was three, and all studies performed a search on MEDLINE (PubMed). The second most searched database was Cochrane (*n* = 10), followed by Embase and SPORTDiscus (*n* = 9), Scopus (*n* = 8), PEDro (*n* = 7), and Web of Science (*n* = 6). Other databases were searched, but the number was minimal.

To assess the risk of bias, the Physiotherapy Evidence Database (PEDro) scale was adopted in six studies, while the Cochrane tools were adopted in two studies, such as the Tool for the Assessment of Study Quality and Reporting in Exercise (TESTEX). Other studies adopted different methods, and two studies did not provide this information.

Eleven studies assessed disability with the Expanded Disability Status Scale (EDSS). Five studies did not use the Expanded Disability Status Scale (EDSS). The range of the EDSS varied widely among the studies. The type of MS was very heterogeneous. The different types of MS, including relapsing-remitting, primary progressive, and secondary progressive, were considered as a whole without particular differences within the same systematic review. There were no systematic reviews that included studies with only one type of MS.

Most of the studies evaluated the effects of an intervention on fatigue and postural balance (*n* = 4). Muscle function was investigated three times. Quality of life and walking ability was investigated twice (*n* = 2). Only one time were cardiorespiratory fitness, depressive symptoms, and cognitive performance investigated.

From a physical point of view, exercise training had positive effects on postural balance (*n* = 4), muscle function (*n* = 3), aerobic capacity, walking ability, physical function, functional mobility, strength, general physical performance, flexibility, and core stability (*n* = 1). One study detected mixed results for dynamic balance [35]. One study [36] found that, in the functional reach test, Pilates exercises were effective, but there was no difference with the control group. One study detected no differences in functional mobility and cardiorespiratory fitness [37].

Three studies detected positive effects of training on fatigue. It seems that yoga had only short-term effects on fatigue and mood and temporary benefits for depression. One study detected a lack of effects on depression [37]. Physical training also had positive effects on quality of life (*n* = 4). Training also had positive outcomes on pain. It seems that training did not work for cognitive performance [38]. More details about the study characteristics are provided in Table 1.

### 3.2. Characteristics of the Interventions

The frequency ranged from one to six times a week, with most of the studies having a mean of three times a week. The intensity ranged from low to vigorous, with some studies highlighting that it progressively increased. The time (duration of the session) ranged from 15 to 135 min, with a mean of about 50 min. The duration of the intervention ranged from 3 to 26 weeks. One study highlighted that it was not clear whether exercise frequency and duration/volume modalities positively influenced depressive symptoms [41].

Aerobic and resistance training was proposed in nine studies: In eight of them, the authors investigated both training modalities individually and combined. Pilates and yoga as standalone interventions were proposed in two studies and aquatic therapy and equine-assisted therapy in one study. Studies on aerobic and resistance training were contradictory, with some studies associating the two training modalities and demonstrating a reduction in perceived fatigue [49] and other studies presenting a negative association between aerobic training and fatigue, whereas muscle strength training presented heterogeneity in the results [51]. Yoga seems to have had positive effects on fatigue [48]. A significant improvement in self-perceived fatigue was detected after Pilates interventions [37], hippotherapy [35], and aquatic therapy [47]. Yoga and aerobic training were more effective in improving dynamic and static balance; aquatic and aerobic training were more effective in improving functional walking ability [43]. Resistance training demonstrated the strongest improvement in the 6 min walking test, while combined training showed the greatest improvement in walking endurance [46]. Another study detected positive effects of Pilates training on postural balance, but it was comparable to aerobic and traditional exercises [36].

Nine studies did not report information on whether the intervention was supervised or not, while the remaining study detected both supervised and home-based methodologies, making it difficult to extrapolate a clear result from the intervention mode.

Related to dropout rate, four studies reported percentages that ranged from 0 to maxima of 18.4% [44], 22% [41], 32% [39], and 47% [42]. More details about the study characteristics are provided in Table 2.

### 3.3. Risk of Bias Assessment

The quality of the included studies ranged from 4 to 10, with a mean of 8/11. Within the included studies, the overall quality was mainly moderate and the risk of bias was medium. Four studies had no scores. The results of the two reviews were unclear. A summary is provided in Table 1 and Table 3.

## 4. Discussion

The main finding of the study was the heterogeneity that still exists in the participants’ characteristics and the protocols adopted in the studies that wanted to associate physical activity intervention and MS. Our findings are in line with another review of reviews [52], which highlighted the heterogeneity in the results with different types/modes of exercise interventions, comparison groups, and/or study populations. Even though several years had passed between the two works, as the other review of reviews was written in 2017 [52], it was also for us difficult to provide information about which specific combination of exercise duration, frequency, and intensities can be suggested. Despite the heterogeneity of the studies, one kind of activity adopted in most of the interventions was mindfulness activities such as yoga and Pilates, which seem safe and feasible, making them ideal as basic interventions. Below is a possible training intervention that should be personalized according to the characteristics of the participants.

A comparison of people with MS with different ages or different disease courses was hard if not impossible to execute. Most of the studies correctly evaluated their samples with the EDSS, which rates the central nervous system’s functioning and defines the development of the disease [53]. Unfortunately, not all studies adopted this scale, and due to the differences within and between the studies, it was hard to create groups. This limitation was also noted in other studies [54,55]. Based on these findings, it is advisable for future studies to be more consistent about this aspect. Only this way will it be possible to create standardized protocols that physicians and kinesiologists can adopt as an intervention. The second limitation is in the protocols adopted. The differences were too broad to make the protocols comparable and generalize the findings. The differences between the studies and the necessity of well-designed interventions were also noted in other studies within the reviews [56,57], making it difficult to synthesize the results [49]. Despite these significant limitations, most of the studies agreed that physical activity is a feasible, cheap, and easy-to-adopt intervention to improve the physical sphere of people with MS (see Table 1). In summary, exercise training has positive effects on postural balance [58], muscle function, aerobic capacity [45], walking ability, physical function, functional mobility, strength [44], general physical performance, flexibility core stability [59], and fatigue [60], cognition [61], depression, pain, and on quality of life [62]. It seems to have effects on symptoms of fatigue, poor functionality, postural balance, and quality of life [63] as well as on chronic levels of BDNF [56,64].

The following is the rationale for the training protocol proposal in terms of frequency, intensity, time, and type. It is fundamental to highlight that the proposal has to be personalized according to the indication of the family doctor or the expert who follows the patient from a medical point of view. The frequency in the studies ranged from one to six times a week, but most of the studies based their weekly frequency on three times a week, making this a reference point. The intensity should be adapted to the patient, but, as suggested in some of the included studies, it should be gradually increased to moments of vigorous intensity. Also, the time ranged significantly, from a few minutes to hours, but the mean was about 50 min, making it an appropriate indication of good training. Lastly, the type of activity: Aerobic and resistance training were proposed in most of the studies (see Table 2), including combined and in the water, and mindfulness activities were also included. According to the included studies, it seems that more physically oriented training (aerobic and resistance training) affects the body characteristics, whereas mindfulness activities have positive effects on fatigue and depression. Considering the importance of muscle strength in older adults and considering the age that current PwMS have reached, it is fundamental to include in the intervention strength-based resistance training such as high-intensity interval exercise (HIIT) to improve strength, gait speed, and quality of life [65] and to prevent sarcopenia [66].

An important aspect to consider in the interventions is the wish to change behavior [67]. Regarding the location, home-based or supervised were both proposed without distinction. Because not all people with MS can constantly go to intervention centers, technology-based distance physical rehabilitation interventions could be a good solution with positive outcomes [68]. Furthermore, home-based exercises are potentially able to reduce fall outcomes in ambulatory PwMS [69], so this training modality can be proposed. This standard operating procedure is a proposal, but future studies on exercise training should deeply standardize their procedures; indeed, this type of intervention has positive outcomes without prescription as a pharmacotherapy requirement [70]. Despite the training typology or structure, physical activity should be suggested because of its benefits and also considering the low dropout rate detected in the included studies (Table 2).

This study is a review of reviews, and its main limitation is that it reports findings that were extrapolated and interpreted from other studies. Despite the effort to minimize the risk of bias and include only high-quality research, some of the findings could have presented errors due to this double indirect topic evaluation. Another significant limitation is related to the impossibility of analyzing the pharmaceutical medication of the patients and the associated physical activity dose. Despite that, this work provides an overview of the topic, providing some feedback for future studies. In the next few years, scientists should try to propose standardized and validated interventions based on a person’s macro-characteristics to prescribe physical activity like medicine.

## 5. Conclusions

The study highlights the necessity of well-planned and structured interventions with standardized protocols proposed in a similar population with multiple sclerosis. These protocols can be adapted and integrated with the medical doctor’s indications to make them personalized to the person’s characteristics and necessities. The protocols should be validated and standardized in order to prescribe them as medicine. Physical activity is a feasible, cheap, and easy-to-adopt intervention to improve the physical, mental, and social health of people with MS.

## Figures and Tables

**Figure 1 jfmk-08-00154-f001:**
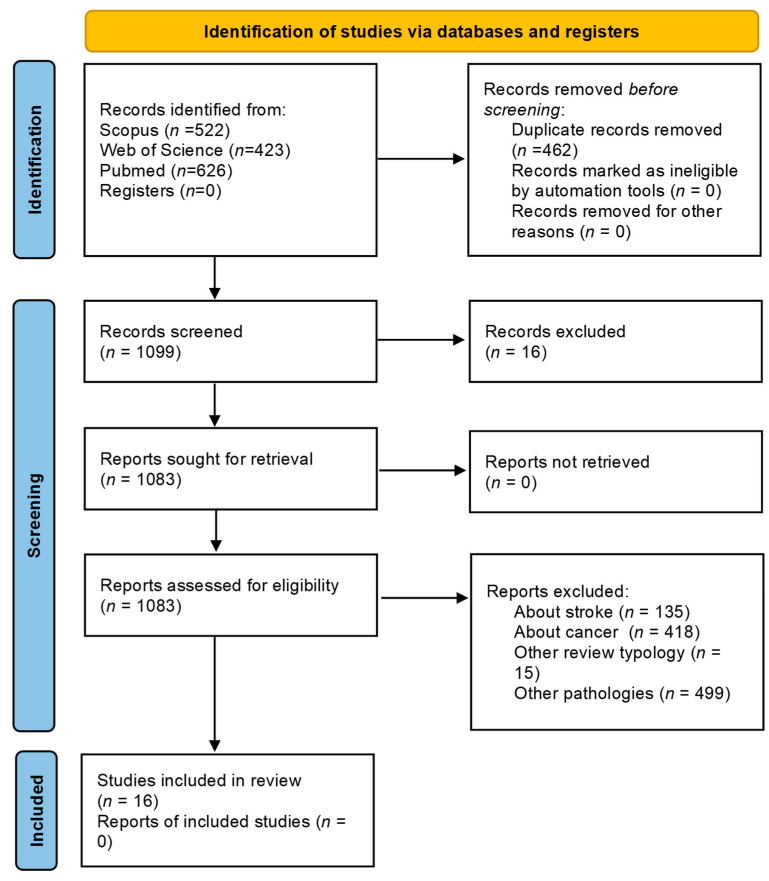
Flow diagram of the study identification.

**Table 1 jfmk-08-00154-t001:** Characteristics of the included studies.

1st Author, Year	Guideline	Databases Searched	Main Objective	No. of Study	Risk of Bias	Main Conclusions
Afkar [39]	NI	MEDLINE, Scopus, Google	Study the effect size of exercise therapy on pwMS’ quality of life in physical and mental dimensions	31	Quality range from 4 to 8	High quality of life was determined for 12 vs. 8 weeks of exercise and was found to be lower
Cramer, 2014 [40]	PRISMA	MEDLINE, Scopus, Cochrane	Examine accessible information on yoga’s efficiency and safety	9	Cochrane tool: overall medium	There were short-term effects of yoga on fatigue and mood but not on health-related quality of life or mobility. No evidence was found for the effects of yoga compared to exercise.
Dalgas [41]	PRISMA	MEDLINE, Embase, Cochrane, PEDro, SPORTDiscus	Study the effects of exercise on depressive symptoms in pwMS	12	PEDro score: 5.6–1.3 points	There was temporary improvement after 5 and 10 weeks of intervention but not after 15 weeks
Dennett 2020 [42]	PRISMA	MEDLINE, Scopus, Embase, PEDro, SPORTDiscus, WoS	Summarize interventions and identify moderators related to adherence and dropout	93	TESTEX rating scale: 7.5/15	Half of the existing exercises reported data on adherence and dropout.
Gharakhanlou, 2021 [38]	PRISMA	MEDLINE, Embase, Cochrane, SPORTDiscus	Investigate how exercise affects pwMS’ overall cognitive performance	13	TESTEX: non-sufficient	Exercise training did not have significant effects on global cognitive performance, attention, executive function, or learning/memory.
Hao [43]	NI	MEDLINE, Embase, Cochrane, WoS, CNKI	Study the effects of 7 different exercise therapies on the balance function and functional walking ability of pwMS	31	13% high risk of bias	Exercise interventions improved dynamic and static balance and the functional walking ability of pwMS.
Isintas Arik, 2022 [36]	PRISMA	MEDLINE, Scopus, PEDro, Science Direct	Evaluate the effects of Pilates workouts on balance in pwMS	8	PEDro results ≥ 4	Pilates improved static and dynamic balance.
Jørgensen, 2017 [44]	PRISMA	MEDLINE, Embase, Cochrane, SPORTDiscus, PEDro	Association of how progressive resistance training affects muscle function	10	PEDro score ≥ 5	Progressive resistance training increased muscle strength, muscle power, and explosive muscle strength.
Langeskov-Christensen [45]	PRISMA	MEDLINE, Embase, Cochrane, PEDro, CINAHL, SPORTDiscus	Evaluate the effects of longitudinal studies evaluating training-induced effects on VO2max in PwMS	17	PEDro score: 5.5 ± 1.5 (range 3–8)	Aerobic training improved aerobic capacity to a level close to the reduction of secondary health risks.
Pearson 2015 [46]	PRISMA	MEDLINE, Embase, Cochrane, CINAHL, SPORTDiscus	Provide information on exercise for improving walking ability in pwMS	13	PEDro scale ≥ 6	Improvement in walking ability was seen.
Sánchez-Lastra, 2019 [37]	PRISMA	MEDLINE, Scopus, PEDro, SPORTDiscus	Study the effects of Pilates on pwMS	14	NI	There were significant effects on quality of life, pain, walking ability, and physical function but not on functional mobility, cardiorespiratory fitness, or depression.
Shariat, 2022 [47]	PRISMA	MEDLINE, Cochrane, WoS, Ovid	Evaluate the effects of long-term aquatic training on balance, fatigue, and motor function in pwMS	16	Joanna Briggs checklists: above 8	Aquatic therapy improved physical fatigue to a greater extent than the control group; it significantly improved fatigue and balance.
Shohani, 2020 [48]	PRISMA	MEDLINE, Scopus, Cochrane, WoS, Science Direct	Investigate how yoga impacts pwMS’ quality of life and level of fatigue	10	High or uncertain risk of bias	Yoga was not healing for fatigue or health-related quality of life.
Suarez-Iglesias, 2021 [35]	PRISMA	MEDLINE, Scopus, PEDro, SPORTDiscus	Assess the information on the potential healing properties of EAT in PwMS	9	PEDro results ≥ 6	There were significant effects on static balance, walking distance, quality of life, spasticity, pain, and incontinence. No significant results were found for depression and constipation or muscular strength.
Taul-Madsen, 2021 [49]	PRISMA	MEDLINE, Scopus, Embase, WoS, SPORTDiscus	Study the effects of exercises on lower limb function and perceived fatigue in pwMS	27	NI	Resistance and aerobic training improved lower extremity physical function and perceived fatigue.
Torres-Costoso [50]	PRISMA	MEDLINE, Embase, Cochrane, WoS. SPORTDiscus	Determine whether and which physical exercise has a positive influence on fatigue	58	RoB2: low risk of bias	Physical exercise reduced fatigue.

Note: Cochrane Collaboration’s tool for assessing the risk of bias: RoB2; equine assistant therapy: EAT; person with multiple sclerosis: pwMS; Tool for the Assessment of Study Quality and Reporting in Exercise: TESTEX; Web of Science: WoS.

**Table 2 jfmk-08-00154-t002:** Characteristics of the interventions.

1st Author, Year	Number of Participants, MS Type	Intervention	Main Results
Afkar [39]	No.: 535, not reported	Mixed	D: 4–12 weeks; F: 2–3/week; D/s: 20–75 min; aerobic, yoga, combination, aquatic, and resistance
Cramer, 2014 [40]	No.: range of 20–314, mixed	Yoga	D: 8–24 weeks; F: 1–3/week; D/s: 60–90 min; Hatha yoga, Iyengar yoga, yoga postures and meditation or relaxation, and yogic breathing techniques.
Dalgas, 2015 [41]	No.: 591, mixed	Mixed	D: 3–26 weeks; endurance training, resistance training, combined training, or other exercise modalities, including sports climbing, yoga, and water activities. Three studies evaluated several exercise interventions.
Dennett, 2020 [42]	No.: 4007, mixed	Mixed	D: 3–26 weeks; F: 1–7/week; yoga, cycling, body weight, mobility, strength training, balance training, endurance, and stretching
Gharakhanlou, 2021 [38]	No.: 639, mixed	Aerobic training and resistance training	D: 8–26 weeks; F: 2–4/week; D/s: 20–60 min I: from low to vigorous, with different modalities of intervention; aerobic, resistance exercises, or mixed; 1 added balance
Hao, 2022 [43]	No.: 904, not reported	Mixed	D: 2–24 weeks; F: 1–6/week; D/s: 10–60 min; aquatic, yoga, Pilates, aquatic, aerobic, resistance, and virtual reality training
Isintas Arik, 2022 [36]	No.: 349, mixed	Pilates	D: 8–12 weeks; D/s: 45–60 min; I: low to moderate; F: 1–3/week
Jørgensen, 2017 [44]	No.: 236, not reported	Progressive resistance training	D: 3–24 weeks; F: 2–5/week; I: progressively increasing between 50 and 90% of 1RM. Number of exercises from 2 to 5; one study included upper body exercises.
Langeskov-Christensen [45]	No.: 330, not reported	Aerobic	D: 3–26 weeks, F: 2–5/week; D/s: 15–45 min; cycling, treadmill walking, rowing, and aquatic aerobics
Pearson 2015 [46]	No.: range 12–119, not reported	Mixed	D: 4–26 weeks; aerobic, yoga, mixed, resistance, and balance
Sánchez-Lastra, 2019 [37]	No.: 507, mixed	Pilates	D: 8–16 weeks; F: 1–3/week; D/s: 15–90 min; I: controlled using the color of the TheraBand^®^
Shariat, 2022 [47]	No.: 794, mixed	Aquatic therapy	D: 3–20 weeks; D/s: 45–135 min; freestyle swimming and shallow water calisthenics aerobics exercise, Ai-Chi exercise in the swimming pool, ergometer water group, and aquatic plyometric exercises
Shohani, 2020 [48]	No.: 693, not reported	Yoga	D: 8–24 weeks; F:1–3/week; D/s: 60–120 min each; Hatha yoga and Iyengar yoga
Suarez-Iglesias, 2021 [35]	No.: 225, not reported	Equine-assisted therapy	D: 8–24 weeks; F: 1–2/week; D: 20–50 D/s
Taul-Madsen, 2021 [49]	No.: 966, mixed	Aerobic training and resistance training	AT. D: 3–26 weeks; F: 1–5/week, D/s: 27–69 min; I: moderate, high, or unknown RT. D: 8–24 weeks; F: 1–3/week; D/s: 30–60 min; I moderate, high, or unknown
Torres-Costoso, 2022 [50]	No.: 2644, not reported	Mixed	D: 4–26 weeks; F: 1–5/week; D/s: 15–120 min; aerobic, stretching, flexion and rotation movements, resistance, combined, yoga, Pilates balance, mobilization, and aquatic exercise + current treatment.

Note: duration of program: D; frequency: F; intensity: I; duration of each session D/s; number: No.; multiple sclerosis: MS; aerobic training: AT; resistance training: RT.

**Table 3 jfmk-08-00154-t003:** Quality assessment through the Assessment of Multiple Systematic Reviews” (AMSTAR) of the included systematic reviews.

1st Author, Year	1	2	3	4	5	6	7	8	9	10	11	Total
Afkar [39]	0	1	1	0	0	1	1	0	1	1	1	7
Cramer, 2014 [40]	0	1	1	1	1	1	1	0	1	1	1	9
Dalgas, 2015 [41]	0	1	1	0	1	1	1	1	1	1	1	9
Dennett, 2020 [42]	1	1	1	0	0	1	1	0	1	0	1	7
Gharakhanlou, 2021 [38]	1	1	1	1	1	1	1	0	1	1	1	10
Hao, 2022 [43]	0	1	1	0	0	1	1	0	1	0	1	6
Isintas Arik, 2022 [36]	1	1	1	1	1	1	1	1	0	0	1	9
Jørgensen, 2017 [44]	0	1	1	1	1	0	1	0	1	0	1	7
Langeskov-Christensen [45]	0	1	1	0	0	1	1	0	1	0	0	5
Pearson, 2015 [46]	0	1	1	0	0	1	1	0	1	0	0	5
Sánchez-Lastra, 2019 [37]	1	0	1	1	1	1	1	1	1	0	1	9
Shariat, 2022 [47]	0	1	1	1	1	1	1	1	1	0	1	9
Shohani, 2020 [48]	1	1	1	0	1	1	1	1	1	1	1	10
Suarez-Iglesias, 2021 [35]	0	1	1	1	1	1	1	0	1	1	1	9
Taul-Madsen, 2021 [49]	1	1	1	1	1	1	1	0	1	0	0	8
Torres-Costoso, 2022 [50]	1	1	1	0	1	1	1	0	0	1	0	7

Note: (1) Was an a priori design provided? (2) Was there duplicate study selection and data extraction? (3) Was a comprehensive literature search performed? At least two electronic sources include years and databases used (e.g, Central, EMBASE, and MEDLINE) (4) Was the status of publication (i.e., gray literature) used as an inclusion criterion? (5) Was a list of studies (included and excluded) provided? (6) Were the characteristics of the included studies provided? (7) Was the scientific quality of the included studies assessed and documented? (8) Was the scientific quality of the included studies used appropriately in formulating conclusions? (9) Were the methods used to combine the findings of studies appropriate? (10) Was the likelihood of publication bias assessed? (11) Were potential conflicts of interest included?

## Data Availability

All data are included in the tables.

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
