# Peer review of "An Overview of Physical Exercise Program Protocols and Effects on the Physical Function in Multiple Sclerosis: An Umbrella Review"

_jfmk, 2023, doi:10.3390/jfmk8040154_

Round 1

Reviewer 1 Report

Comments and Suggestions for Authors The need for individualized physical exercise is not mentioned and articles targeting this area should be included as pharmaceutical medication is now aimed at individualized treatment.
Comments on the Quality of English Language line 31: immune-mediated instead of autoimmune
line 61: decrease instead of contrast

Author Response

The need for individualized physical exercise is not mentioned and articles targeting this area should be included as pharmaceutical medication is now aimed at individualized treatment.

Reply: Thank you for the comment, we totally agree with the Reviewer indication. Unfortunately, it has been impossible to extrapolate pharmaceutical medication (such as information related to the sclerosis multiple stage) data from the included.

Otherwise, we added some sentence to highlight the importance of a individualized intervention: Line 244-245, line 267-269, Line 306-308

Furthermore, in the limit of the study we highlighted the impossibility to collect this type of data (Line 298-300).

Comments on the Quality of English Language

line 31: immune-mediated instead of autoimmune
line 61: decrease instead of contrast

Reply: Thank you for the comments. We corrected the words directly in the manuscript

Reviewer 2 Report

Comments and Suggestions for Authors

I think the proposed review is interesting, as it focuses on the opportunity to propose physical activity even in a pathological condition, I have some suggestions to improve the work:

- It would be interesting to have an opinion from the authors on what the best types of training could be, in terms of effectiveness and also "safety" for example in order not to accentuate an inflammatory state, which is probably present, but also managing to provide an adequate training stimulus

- You will distinguish pilates well from others, I would not have considered hippotherapy

- I would give more emphasis to strength training which in fact proves to be the most effective, perhaps proposing schemes to be adopted such as suspension training (10.1055/a-0787-1548, 10.3389/fspor.2022.950949, 10.1016/j.ghir.2020.101320 ) or propose low-intensity schemes

- are there any strength assessments? Dynamometer, platform...

Comments on the Quality of English Language

It needs some revisions

Author Response

I think the proposed review is interesting, as it focuses on the opportunity to propose physical activity even in a pathological condition, I have some suggestions to improve the work:

It would be interesting to have an opinion from the authors on what the best types of training could be, in terms of effectiveness and also "safety" for example in order not to accentuate an inflammatory state, which is probably present, but also managing to provide an adequate training stimulus

Reply: Thank you for the suggestion, we added a sentences in the first part of the discussion (line 241-244) to provide a suggestion based on the findings of the study.

You will distinguish pilates well from others, I would not have considered hippotherapy

Reply: Thank you for the comment. You are right about hippotherapy, it is far from the other training typology, but we could not exclude because it felt within the all previously-written eligibility criteria and thus within the PRISMA statement.

I would give more emphasis to strength training which in fact proves to be the most effective, perhaps proposing schemes to be adopted such as suspension training (10.1055/a-0787-1548, 10.3389/fspor.2022.950949, 10.1016/j.ghir.2020.101320 ) or propose low-intensity schemes

Reply: Thank you for the suggestion. We added in the text a sentence about this aspect (line 279-283) adopting two of the three articles suggested by you

Are there any strength assessments? Dynamometer, platform...

Reply: Thank you for the comment. Our interest has been more on the protocols adopted than on the assessments. We took this decision because the articles included in the reviews adopted different type of protocols containing a lot of different direct and indirect method to evaluate the physical (endurance, force and flexibility), social and psychological spheres (mainly questionnaire) making impossible to analyze and summarize it properly.

English: It needs some revisions

Reply: The manuscript has been double checked for the English. Some errors have been corrected and sentences improved. Thank you for the suggestion

Reviewer 3 Report

Comments and Suggestions for Authors

First and foremost, I'd like to commend the authors on the work they have done. After carefully reading the text and references, I have only one point of concern to raise: Reference [37] does not appear to demonstrate effects on depression; in fact, I believe it shows a lack of effects. Apart from this necessary point for consideration, I have nothing else to report.

Author Response

First and foremost, I'd like to commend the authors on the work they have done. After carefully reading the text and references, I have only one point of concern to raise: Reference [37] does not appear to demonstrate effects on depression; in fact, I believe it shows a lack of effects. Apart from this necessary point for consideration, I have nothing else to report.

Reply: Thank you for the time spent on revisioning the manuscript. Thank you for the correction, it has been an error in the text. We corrected it (line 178). We double checked the table and it seems correct on the table.
